# Postnatal Development of Synaptic Plasticity at Hippocampal CA1 Synapses: Correlation of Learning Performance with Pathway-Specific Plasticity

**DOI:** 10.3390/brainsci14040382

**Published:** 2024-04-14

**Authors:** Yuheng Yang, Yuya Sakimoto, Dai Mitsushima

**Affiliations:** 1Department of Physiology, Yamaguchi University Graduate School of Medicine, Yamaguchi 755-8505, Japan; olliveryang@gmail.com (Y.Y.); ysaki@yamaguchi-u.ac.jp (Y.S.); 2The Research Institute for Time Studies, Yamaguchi University, Yamaguchi 753-8511, Japan

**Keywords:** contextual learning, synaptic diversity, information entropy, developmental critical periods, AMPA receptor, GABA_A_ receptor

## Abstract

To determine the critical timing for learning and the associated synaptic plasticity, we analyzed developmental changes in learning together with training-induced plasticity. Rats were subjected to an inhibitory avoidance (IA) task prior to weaning. While IA training did not alter latency at postnatal day (PN) 16, there was a significant increase in latency from PN 17, indicating a critical day for IA learning between PN 16 and 17. One hour after training, acute hippocampal slices were prepared for whole-cell patch clamp analysis following the retrieval test. In the presence of tetrodotoxin (0.5 µM), miniature excitatory postsynaptic currents (mEPSCs) and inhibitory postsynaptic currents (mIPSCs) were sequentially recorded from the same CA1 neuron. Although no changes in the amplitude of mEPSCs or mIPSCs were observed at PN 16 and 21, significant increases in both excitatory and inhibitory currents were observed at PN 23, suggesting a specific critical day for training-induced plasticity between PN 21 and 23. Training also increased the diversity of postsynaptic currents at PN 23 but not at PN 16 and 21, demonstrating a critical day for training-induced increase in the information entropy of CA1 neurons. Finally, we analyzed the plasticity at entorhinal cortex layer III (ECIII)-CA1 or CA3-CA1 synapses for each individual rat. At either ECIII-CA1 or CA3-CA1 synapses, a significant correlation between mean α-amino-3-hydroxy-5-methyl-4-isoxazole propionic acid/N-methyl-D-aspartic acid (AMPA/NMDA) ratio and learning outcomes emerged at PN 23 at both synapses, demonstrating a critical timing for the direct link between AMPA receptor-mediated synaptic plasticity and learning efficacy. Here, we identified multiple critical periods with respect to training-induced synaptic plasticity and delineated developmental trajectories of learning mechanisms at hippocampal CA1 synapses.

## 1. Introduction

The hippocampus is an essential center for cognitive processes related to learning and memory, and it has a crucial role in the creation of episodic memories [1], the encoding of spatio-temporal information related to specific events [2,3,4,5], and the consolidation of experiences into a coherent memory framework. The hippocampus, however, undergoes significant developmental changes during the postnatal period, and these changes profoundly impact its functional capabilities [6,7,8,9].

In the developing rat hippocampus, synaptic contacts in CA1 neurons exhibit continuous growth during the initial weeks of life [10,11,12]. Spine density notably increases between postnatal day (PN) 7 and 28, with the most significant growth occurring between PN 7 and 21, resulting in a threefold expansion [13]. Furthermore, the intensity of long-term potentiation (LTP), a fundamental mechanism underlying learning and memory, substantially rises from PN 10 to 20, persisting until PN 60 [14]. The latest research on comprehensive synapse analysis in rats throughout their lifespan highlights a remarkable increase in the molecular and morphological diversity of excitatory synapses during the first postnatal month, particularly within the hippocampus [15]. In recent studies using mouse models, it observed that the immature hippocampus lacks a competitive neuronal engram allocation process, which, in turn, impedes the formation of sparse engrams and the development of precise memories. This developmental restriction persists until approximately the fourth postnatal week in mice when inhibitory circuits within the hippocampus reach a critical level of maturation [16].

While these findings have shed light on the developmental processes of memory in the hippocampus, it is important to note that the hippocampus is not a standalone structure. CA1 neurons within the dorsal hippocampus play a fundamental role in encoding experienced episodes [5], processing information from the entorhinal cortex (EC), and mediating distinct synaptic pathways. Specifically, direct temporoammonic projections originating from EC layer III neurons, known as the TA pathway, contrast with EC layer II neurons that project indirectly through the dentate gyrus (DG) and CA3 regions, forming the trisynaptic pathway [17,18]. Both pathways converge on individual CA1 pyramidal neurons, with CA1 dendrites establishing synapses with the TA pathway in the stratum lacunosum moleculare and with Schaffer’s collaterals from CA3 in the stratum radiatum [19]. These specific pathways contribute to the complex network involved in memory formation and retrieval. These interactions between the EC, CA3, and CA1 are critical for the processing of spatial and temporal information related to specific events, which are fundamental aspects of episodic memory [20]. We revealed how contextual learning abilities change across different postnatal stages and explored the correlation between these abilities and pathway-specific CA1 synaptic plasticity in the hippocampus.

## 2. Materials and Methods

### 2.1. Animals

This study utilized male, pre-weaned Sprague-Dawley rats (Table 1). All rats were housed with the mother rat and siblings in opaque plastic cages lined with wood chips (height 25 cm; length 25 cm; width 40 cm) at a constant stable temperature (24 ± 1 °C) under a 12 h light/dark cycle (lights on 8:00 a.m. to 8:00 p.m.) with ad libitum access to water and food (MF, Oriental Yeast Co. Ltd., Tokyo, Japan). All animal housing and surgical procedures followed the Institutional Animal Care and Use Committee guidelines of Yamaguchi University. All experimental procedures were approved by the Institutional Animal Care and Use Committee of Yamaguchi University Graduate School of Medicine (Approval No. 04-S02). These guidelines comply with the Guide for the Care and Use of Laboratory Animals published by the National Institutes of Health (NIH Publication No. 85-23, revised 1996).

### 2.2. Inhibitory Avoidance (IA) Task

We described hippocampus-dependent IA training procedures previously [21,22]. The IA training apparatus (height: 33 cm; length: 33 cm; width: 58 cm) consisted of a two-chambered box that contained a lighted safe side and a dark shock side. The chambers were divided by a trap door (Figure 1A). For training, rats were initially placed in the well-lit section of the box, positioned facing the corner opposite the entrance. Once the trap door was opened, the rats had the freedom to enter the adjacent dark compartment. We measured the latency as a behavioral parameter before the rats entered the novel dark area (referred to as latency before IA learning, as shown in Figure 1C). Upon entry into the dark side, we promptly closed the door and administered a brief scrambled electrical foot shock (1.6 mA, 2 s) through electrified steel rods embedded in the floor of the box. After enduring the dark compartment for 10 s, the rats were returned to their respective home cages. Untrained control rats remained undisturbed in their home cages.

Thirty minutes after the above procedure, rats were placed on the lit side to test their memory retrieval. The latency to enter the dark box was assessed as an indicator of learning performance (latency after IA training, Figure 1C). Rats were euthanized 30 min after the retrieval test.

### 2.3. Electrophysiological Recording of Slice-Patch Clamping

We have previously published a detailed technical protocol for the slice-patch clamp technique used to analyze training-induced synaptic plasticity, along with a brief demonstration movie [22,23]. In brief, one hour after delivering the paired foot shocks, rats were anesthetized with pentobarbital, and acute brain slices were prepared [21]. Naive rats were used as the untrained control group, and they all received the same dose of anesthesia in the home cages. In order to carry out whole-cell recordings, the brains were quickly perfused with ice-cold dissection buffer (containing 25.0 mM NaHCO_3_, 1.25 mM NaH_2_PO_4_, 2.5 mM KCl, 0.5 mM CaCl_2_, 7.0 mM MgCl_2_, 25.0 mM glucose, 90 mM choline chloride, 11.6 mM ascorbic acid, and 3.1 mM pyruvic acid) and gassed with a mixture of 5% CO_2_ and 95% O_2_. Coronal brain sections (targeting the CA1 area at AP −3.8 mm, DV 2.5 mm, and LM ±2.0 mm) were prepared at a thickness of 350 μm using a Leica vibratome (VT-1200; Leica Biosystems, Nussloch, Germany) in dissection buffer. These sections were then transferred to a physiological solution (at 22–25 °C), which consisted of 114.6 mM NaCl, 2.5 mM KCl, 26 mM NaHCO_3_, 1 mM NaH_2_PO_4_, 10 mM glucose, 4 mM MgCl_2_, and 4 mM CaCl_2_ at a pH of 7.4, and were gassed with 5% CO_2_ and 95% O_2_. The recording chamber was consistently perfused with the physiological solution, maintaining a temperature between 22 and 25 °C.

Glass patch-recording pipettes with resistances ranging from 4 to 7 megaohms were meticulously fashioned using a horizontal puller (Model P97; Sutter Instrument, Novato, CA, USA). These pipettes were then filled with an intracellular solution comprising 115 mM cesium methanesulfonate, 20 mM CsCl, 10 mM HEPES, 2.5 mM MgCl_2_, 4 mM Na_2_ATP, 0.4 mM Na_3_GTP, 10 mM sodium phosphocreatine, and 0.6 mM EGTA, adjusted to pH 7.25. To maintain a stable recording environment, artificial cerebrospinal fluid containing 0.1 mM picrotoxin and 4 mM 2-chloroadenosine was continuously perfused through the recording chamber, maintaining a temperature range of 22–25 °C. It is worth noting that previous investigations confirmed the absence of time-dependent changes in plasticity between 1 to 10 h following slice preparation [22], and, therefore, neuronal recordings were conducted within this time frame.

To assess the functionality of CA3-CA1 synapses, a bipolar tungsten stimulating electrode (Unique Medical Co., Ltd., Tokyo, Japan) was meticulously positioned within the stratum radiatum layer of the CA1 region, at an approximate lateral distance of 200–300 μm relative to the recorded cells. Similarly, for ECIII-CA1 synapses, the stimulating electrode was precisely placed within the stratum lacunosum moleculare layer of the CA1 region, approximately 200–300 μm medial to the recorded cells. CA1 pyramidal neurons from the rat hippocampus were subjected to whole-cell recordings using an Axopatch 700 A amplifier (Molecular Devices Inc., San Jose, CA, USA). The complete collection of patch-clamp data was carefully obtained using a Clampex 10.4 instrument (Molecular Devices) and then analyzed using the Clampfit 10.4 software (Molecular Devices).

To evaluate postsynaptic plasticity, we computed the ratio between AMPA receptor-mediated postsynaptic current and NMDA receptor-mediated current (AMPA/NMDA). This ratio was determined by measuring the peak current at −60 mV divided by the current at +40 mV, recorded at 150 ms following stimulus onset (an average of 40–60 traces was compiled for each holding potential). To graphically represent the distribution of the AMPA/NMDA current ratio, we plotted cell-specific ratios on the X-axis and their cumulative frequencies on the Y-axis.

### 2.4. Miniature Postsynaptic Current Recordings

In previous studies, we have provided a comprehensive description of the technical approach used to record miniature synaptic currents [21,24]. mEPSCs are generally thought to be triggered by the release of a single glutamate vesicle from the presynaptic site. In contrast, mIPSCs are thought to originate from GABA release. An increase in the amplitude of mEPSCs and mIPSCs reflects a strengthening of postsynaptic transmission, while an increase in event frequency indicates a greater number of functioning synapses or an enhanced probability of presynaptic release.

For these miniature recordings, we introduced a sodium channel blocker (0.5 μM tetrodotoxin, TTX; Wako Pure Chemical Industries Ltd., Osaka, Japan) into the physiological solution. mEPSCs (at a holding potential of −60 mV) and mIPSCs (at a holding potential of 0 mV) were recorded sequentially from the same CA1 neuron for over a period of 5 min each. Using Clampfit 10.4 software (Molecular Devices), we identified miniature events, focusing on those exceeding 10 pA for analysis. We conducted recordings for a minimum duration of 5 min in order to ascertain the occurrence rate of mEPSCs or mIPSCs. Event amplitudes were averaged to calculate the mean amplitude.

### 2.5. Statistical Analysis

Data and statistical analyses were performed using GraphPad Prism software (Version 10.2.0; GraphPad Software LLC, San Diego, California USA) and standard spreadsheet software (Excel 2010, Microsoft Co., Redmond, WA, USA). Due to the large within-group variability in self-entropy data, log (1 + x) transformation was performed prior to analysis [25]. Unpaired *t*-tests were used to analyze AMPA/NMDA ratio, latency, and postnatal age. To examine the relationship between latency and AMPA/NMDA ratio, we used a linear regression model as the analytical basis. Within this framework, we considered the null hypothesis (H0), which suggested the absence of a significant linear relationship between the two variables, and the alternative hypothesis (Ha), which implied the existence of a notable association. Estimates of the coefficients of the regression model were obtained, and the goodness of fit was evaluated using the squared Pearson correlation coefficient (R²). The statistical significance of both the model and its individual components was determined by examining *p* values. *p* values < 0.05 were considered significant.

## 3. Results

### 3.1. Developmental Change in Contextual Learning Performance

Rats underwent the inhibitory avoidance (IA) task (Figure 1A), where they transitioned from a brightly lit chamber to a dark enclosure, receiving an electric foot shock (1.6 mA, 2 s) upon entry. Thirty minutes post-task completion, we assessed contextual learning performance by measuring the latency exhibited within the illuminated chamber. We recorded the AMPA/NMDA ratios of CA3-CA1 synapses in the dorsal hippocampus (Figure 1B). As an indicator of contextual learning performance, we assessed the latency to enter the dark room before the foot-shock and compared it with the latency 30 min after completing the IA task. Following paired foot-shock exposure, the latency consistently increased after training compared to before training (Figure 1C). When aversive foot-shocks were introduced, the latency consistently increased after PN 17, probably related to the acquisition of a sense of crisis, but no such change was observed at PN 16.

### 3.2. Developmental Change in Training-Induced Diversity at Excitatory and Inhibitory Synapses

The strength of AMPA receptor-mediated excitatory input vs. GABA_A_ receptor-mediated inhibitory input was measured in each neuron and plotted two-dimensionally (amplitude in Figure 2B, upper panel; frequency in Figure 2C, upper panel). Regarding the mEPSCs (Figure 2D), IA training significantly increased their mean amplitude at PN 23 (*p* = 0.0144, unpaired *t*-test) and their frequency at PN 21 (*p* = 0.0112, unpaired *t*-test). In contrast, the mEPSC frequency was significantly reduced at PN 16 (*p* = 0.0064, unpaired *t*-test). Regarding the mIPSCs (Figure 2E), IA training significantly increased their mean amplitude at PN 23 (*p* < 0.0001, unpaired *t*-test). Conversely, mIPSC frequency was significantly decreased at PN 16 (*p* = 0.0006, unpaired *t*-test). However, in order to determine if the plasticity is caused by learning, additional yoked controls exposed to a foot shock (unpaired) or the apparatus alone (walk through) are required [21].

Based on the information theory of Shannon, we further calculated the appearance probability of the mean amplitudes of mEPSCs and mIPSCs (Figure 3A). First, we identified the distribution of the appearance probability in untrained control animals, followed by an analysis of the cell-specific appearance probability of all recorded neurons individually. Each probability of a single neuron was calculated as the self-entropy and plotted two-dimensionally; e.g., a point with a high appearance probability (points close to the mean levels) indicated a low self-entropy, whereas a point with a very rare probability (points away from the mean levels) indicated a high self-entropy. We used a two-dimensional kernel analysis to visualize the diversity of self-entropy (Figure 3A, lower panels).

Similarly, for the mEPSC and mIPSC frequency (Figure 3B), we identified the distribution of appearance probability in untrained controls, followed by the analysis of the appearance probability of all recorded neurons individually. A two-dimensional kernel analysis visualized the diversity of self-entropy (Figure 3B, lower panels). IA training clearly increased the diversity at PN 21 and 23.

Regarding the self-entropy of mEPSC or mIPSC amplitude (Figure 3C), IA training significantly increased it at PN 23, but not at PN 16 and 21 (mEPSC, PN 16: *p* = 0.4965; PN 21: *p* = 0.1073; PN 23: *p* = 0.0484. mIPSC, PN 16: *p* = 0.5524; PN 21: *p* = 0.2269; PN 23: *p* = 0.0002). Regarding the self-entropy of mEPSC frequency (Figure 3D), the training increased it at PN 21 (*p* = 0.0063), but decreased it at PN 16 (*p* = 0.0187).

### 3.3. Training-Induced Plasticity at CA3-CA1 or ECIII-CA1 Synapses

To evaluate training-induced synaptic plasticity specific to CA3-CA1 or ECIII-CA1 synapses, we examined the ratio of AMPA- and NMDA-receptor-mediated currents at these synapses [21]. At CA3-CA1 synapses (Figure 4A), IA training substantially enhanced the AMPA/NMDA ratio at CA3-CA1 synapses during PN 16–23, suggesting the training-induced strengthening at CA3-CA1 synapses. Similarly, at ECIII-CA1 synapses (Figure 4B), IA training substantially increased the ratio at ECIII-CA1 synapses during PN 16–21, suggesting training-induced strengthening at ECIII-CA1 synapses in juvenile rats.

### 3.4. Correlation between Training-Induced Plasticity and Learning Performance

We examined the relationship between contextual learning performance and synaptic plasticity at the CA1 synapses across the different postnatal days. Regarding the CA3-CA1 pathway (Figure 4C), at PN 16–22, there is no discernible linear correlation between the mean AMPA/NMDA ratio and latency, and the relationship lacks statistical significance (*p* > 0.05). However, at PN 23, a positive correlation emerges between the mean ratio at CA3-CA1 synapses and latency (*R* = 0.910), and this correlation was statistically significant (*p* = 0.0313). Similarly, for the ECIII-CA1 pathway (Figure 4D), at PN 16–22, there is no discernible linear correlation between the mean AMPA/NMDA ratio and latency, and the relationship lacks statistical significance (*p* > 0.05). However, at PN 23, a positive correlation emerges between the mean ratio at ECIII-CA1 synapses and latency (*R* = 0.890), and this correlation was statistically significant (*p* = 0.0431). These results indicate that the functional relationship between contextual learning and hippocampal CA1 synaptic plasticity occurs at a specific day of age and is particularly pronounced between PN 22 and 23. This correlation may indicate a functional critical date for contextual learning and plasticity at CA1 synapses.

## 4. Discussion

Here, we focused on pre-weaned animals and examined developmental changes in training-induced synaptic plasticity. At PN 16, but not at PN 17-23, rats failed to maintain a memory of IA learning, suggesting an undeveloped contextual memory function in infant rats [22]. A previous study showed that rats at PN 16 showed basic sensory/motor functions, while they did not avoid aversive pain by becoming immobilized in the light box. Vision arises around PN 13 and 14, and sufficient pain sensitivity developed before PN16 in rats [22]. In addition, movement into the dark box during IA training and the immobilization in the dark box in the light–dark box test suggest sufficient motor function to choose between staying and moving [22]. Although, at PN 16, the infant rats could have decreased IA latency, the memory deficit may not be attributed to the development of sensory/motor functions.

The present study confirmed an increase in the AMPA/NMDA ratio after IA training at many developmental stages. Notably, IA training induced a significant increase in the AMPA/NMDA ratio at CA3-CA1 synapses during PN16-23. This result suggests that contextual learning predominantly strengthened CA3-CA1 synapses in juvenile rats. Similarly, at ECIII-CA1 synapses, IA training significantly increased the AMPA/NMDA ratio during PN 16-21, indicating a predominant strengthening of ECIII-CA1 synapses. These results highlight the dynamic nature of synaptic plasticity, which varies depending on the developmental stage and the specific synaptic pathway.

Our previous study confirmed an increase in the amplitude of AMPA receptor-mediated excitatory postsynaptic currents after IA learning during PN 21-28, particularly finding that the AMPA receptor GluA1 subunit is required for contextual memory. Furthermore, IA training induced an increase in the postsynaptic number of AMPA receptor channels without changing the cation current per single channel in CA1 pyramidal neurons [24]. These results may suggest that the number of postsynaptic AMPA receptors increases even after IA training in early childhood and that AMPA receptors are central to memory formation from PN 17 onward.

A key finding in our study was the emergence of a positive correlation between the AMPA/NMDA ratio and latency at PN 23, for both CA3-CA1 and ECIII-CA1 synapses. This correlation became statistically significant during this developmental stage. This suggests that contextual learning becomes more tightly linked to the strength of hippocampal CA1 synapses, particularly between PN 22 and 23. At PN 16–17, the hippocampus is still undergoing significant developmental changes [26]. While the capacity for learning is heightened, the processes that encode, store, and retrieve learned information may not yet be fully mature [9]. The interplay between NMDA and AMPA receptors is crucial for synaptic plasticity. At around PN 17, the influence of NMDA receptors, which are essential for synaptic plasticity and memory formation, might be more pronounced [27,28,29]. This could potentially overshadow the contributions of AMPA receptors to learning performance, making it difficult to discern clear correlations. By PN 23, the balance between NMDA and AMPA receptor activity may have reached a more stable state, allowing for clearer associations between AMPA receptor-mediated synaptic changes and learning performance to emerge [30,31,32,33]. By PN 23, the neuronal networks and synaptic connections may have matured further, leading to the more efficient and effective processing of learning tasks. This maturation could result in a more pronounced correlation between synaptic changes and learning performance.

Regarding the other evidence of development, naive AMPA receptor densities increased dramatically (184%) during PN 0–10 and were stable during PN 20–30 [34]. We also believe that the number of CA1 neurons and synapses increases until PN 20 and that hippocampal function matures along with it. In addition, the magnitude of the late-phase LTP at PN 12–15 was almost half that at PN 19–35 [35], whereas the magnitude of early-phase LTP increased at PN 15–20 [36,37]. In summary, the hippocampus grows dramatically during the first 4 weeks of life, and the memory system changes accordingly. In particular, at PN 23, as seen in mature rats, contextual experience enhanced excitatory postsynaptic currents of AMPA receptors, but also inhibitory postsynaptic currents mediated by GABA_A_ receptors. In CA1 pyramidal neurons, the ability to form sparse engrams emerges at PN 24 with the maturation of inhibitory circuits in the hippocampus [16].

The observed increase in mE(I)PSC amplitudes at PN 23 is consistent with a previous study highlighting the role of synaptic transmission in learning and memory processes [22]. The decrease in mE(I)PSC frequencies at PN 16 provides an intriguing contrast, suggesting the developmental modulation of synaptic connectivity. It has long been known that LTD tends to be induced in infants [38,39], whereas LTP tends to be induced in juvenile-to-mature rats [40]. The infantile period may represent a phase of synaptic refinement in which neural circuits undergo selective pruning to increase efficiency. The differential effects of IA training on mEPSCs and mIPSCs across developmental stages underscore the complex interplay between excitatory and inhibitory processes in cognitive development.

### Implications and Future Directions

Our study highlights the importance of considering developmental stages when investigating the relationship between synaptic plasticity and learning. Understanding the temporal dynamics of this relationship may have implications for educational and therapeutic approaches targeting memory and learning in developing individuals. Future research should explore the underlying mechanisms that govern this relationship, including the role of specific molecular pathways and cellular processes. In addition, investigating the effects of postnatal synaptic plasticity on cognitive function and memory retention will provide further insight into the complex interactions within the hippocampus during postnatal development.

## 5. Conclusions

Here, we found multiple critical periods for learning and associated plasticity in pre-weaning male rats. Although IA training induced plasticity from PN 16 to 23, functional linkage between learning and plasticity at AMPA receptor-mediated excitatory synapses was observed on PN 23 but not on PN 16–22. IA training also promoted GABA_A_ receptor-mediated postsynaptic current on PN 23 but not on PN 16–22. Identifying the postnatal days critical for learning-dependent synaptic plasticity is essential for the analysis of hippocampal learning function.

## Figures and Tables

**Figure 1 brainsci-14-00382-f001:**
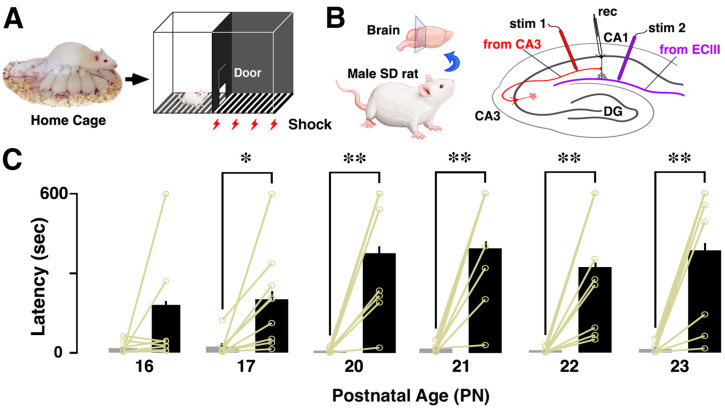
Inhibitory avoidance (IA) task and postnatal development of performance. (**A**) Schematic illustration of the IA training apparatus. (**B**) Diagrams illustrate electrode placements for patch-clamp analysis at CA3-CA1 synapses and ECIII-CA1 synapses. (**C**) Thirty minutes after training, a prolonged latency to enter the dark side of the box was observed from PN 17 to 23. Data are presented as individual data points and expressed as the mean ± SEM. The same rats are connected by yellow lines. * *p* < 0.05, and ** *p* < 0.01 vs. before training.

**Figure 2 brainsci-14-00382-f002:**
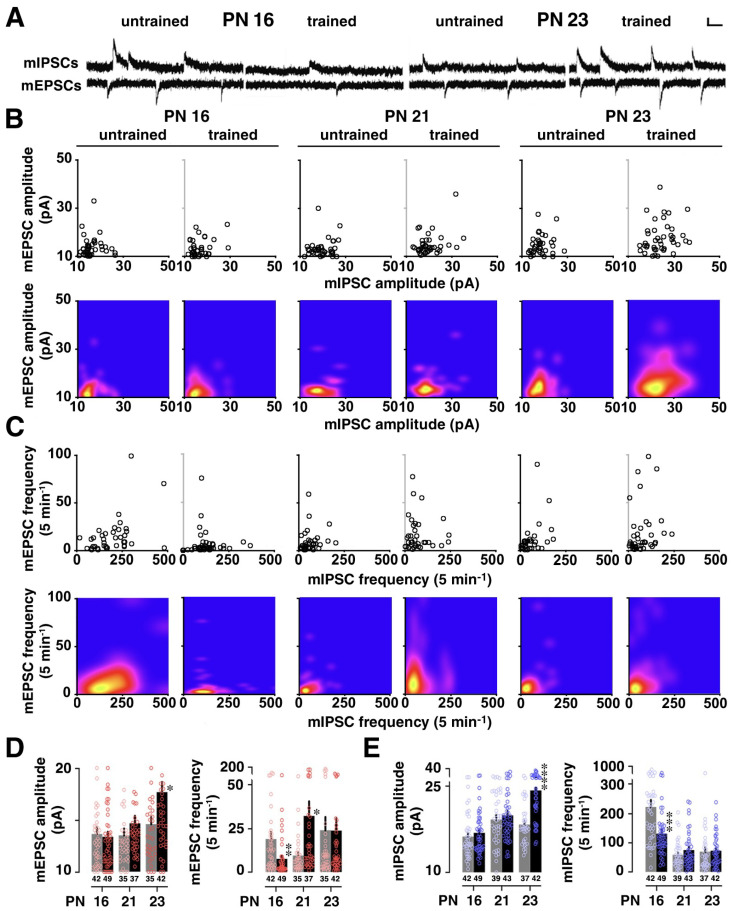
Postnatal development of IA-task-induced synaptic plasticity. (**A**) Representative traces of miniature excitatory postsynaptic currents (mEPSCs) and miniature inhibitory postsynaptic currents (mIPSCs). We recorded the currents sequentially in the same CA1 pyramidal neuron in the presence of tetrodotoxin (0.5 µM). Vertical bar = 20 pA; horizontal bar = 50 ms. (**B**) The upper panel displays two-dimensional plots of the mean mEPSC and mIPSC amplitudes in an individual neuron, and the lower panel presents the kernel density visualizing the distribution of the appearance probability at any point. (**C**) The upper panel displays two-dimensional plots of the mean mEPSC and mIPSC frequencies in an individual neuron, and the lower panel presents the kernel density. (**D**) Postnatal changes in the mean mEPSC amplitude and frequency in untrained (gray) and trained (black) rats. (**E**) Postnatal changes in the mean mIPSC amplitude and frequency in untrained (gray) and trained (black) rats. IA training increased the amplitudes of both excitatory (mEPSCs) and inhibitory (mIPSCs) miniature synaptic events at PN 23, but decreased the frequencies of both mEPSCs and mIPSCs at PN 16. The number of neurons in each group is shown at the bottom of each bar. Error bars indicate + SEM. * *p* < 0.05, ** *p* < 0.01, *** *p* < 0.001, **** *p* < 0.0001 vs. untrained.

**Figure 3 brainsci-14-00382-f003:**
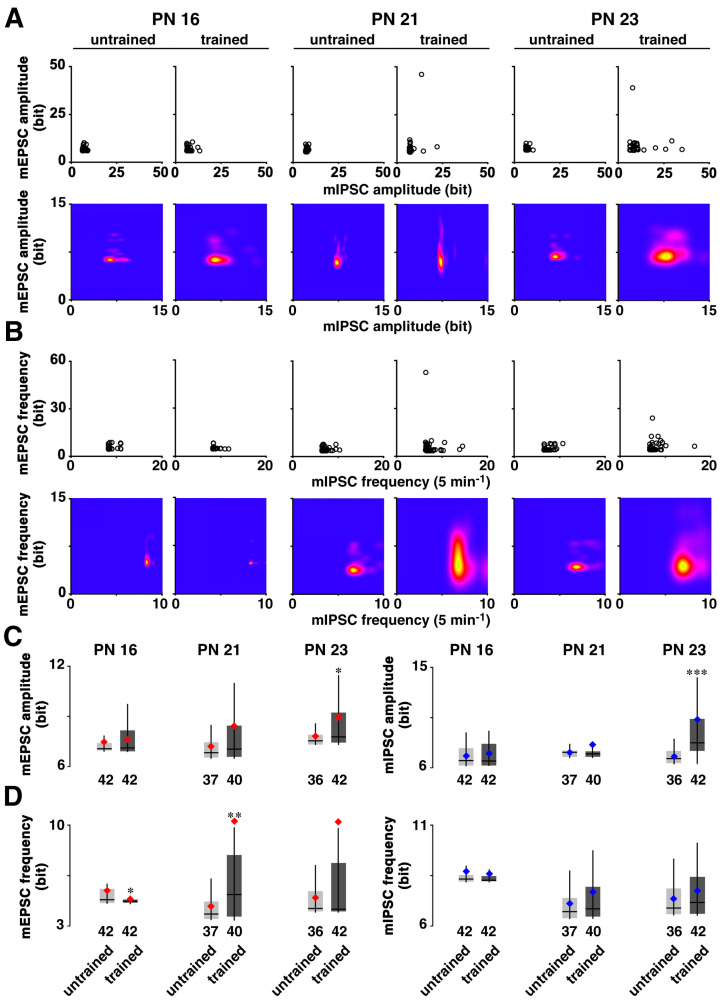
Postnatal development of the gained self-entropy in individual neurons. (**A**) By calculating the appearance probability of individual dots in mEPSC and mIPSC amplitudes, we calculated the self-entropy (bit) of individual neurons (upper), and visualized the kernel density distribution (lower). (**B**) The same process as in (**A**), applied for mEPSC and mIPSC frequencies, showing the self-entropy (upper) and the density distribution (lower). (**C**) Distribution of self-entropy for mEPSC or mIPSC amplitude in individual neurons in PN 16, PN 21, and PN 23 rats. (**D**) Distribution of self-entropy for mEPSC or mIPSC frequency in individual neurons in PN 16, PN 21, and PN 23 rats. The box indicates 50% central area with a line that represents the median. The vertical line indicates minimum to maximum data area without outliers. The red/blue dot represents the average. The number of neurons in each group is shown at the bottom of each bar. * *p* < 0.05, ** *p* < 0.01, *** *p* < 0.001 vs. untrained.

**Figure 4 brainsci-14-00382-f004:**
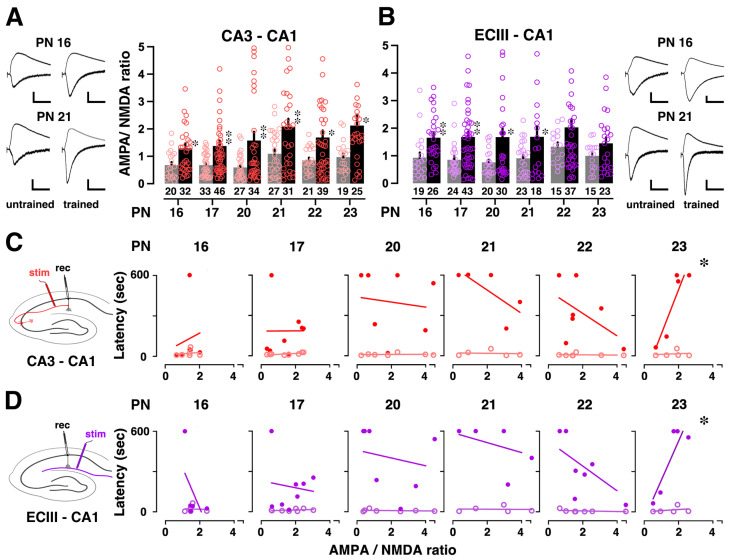
Postsynaptic plasticity at CA3-CA1 or ECIII-CA1 synapses. (**A**) IA training consistently increased the mean AMPA/NMDA ratio at CA3-CA1 synapses after PN 16. The number of cells in each group is shown at the bottom of each bar. The left insets show representative traces. (**B**) IA training increased the mean AMPA/NMDA ratios at ECIII-CA1 synapses, and the number at the bottom of each bar represents the number of cells in each group. Representative traces are shown in the right insets. * *p* < 0.05, ** *p* < 0.01 vs. untrained. (**C**) At the CA3-CA1 synapse, mean AMPA/NMDA ratios for individual animals were plotted against the latency before (open) and after IA training (closed). (**D**) At the ECIII-CA1 synapse, mean AMPA/NMDA ratios for individual animals were plotted against the latency before (open) and after IA training (closed). Significant correlations were observed only for PN 23 animals after training. * *p* < 0.05 by Pearson correlation. Vertical scale bars = 40 pA; horizontal scale bars = 50 ms.

**Table 1 brainsci-14-00382-t001:** Postnatal groups and body weight.

Postnatal Days	Body Weight (g)	Number of Rats
16	36.1 ± 1.3	14
17	38.8 ± 1.4	14
20	45.3 ± 1.8	13
21	49.5 ± 1.6	16
22	51.9 ± 1.6	22
23	55.3 ± 2.4	18

## Data Availability

The entirety of the raw data from this study are available from the authors upon request. The data are not publicly available due to the inclusion of data for ongoing research.

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
