# Peer review of "Postnatal Development of Synaptic Plasticity at Hippocampal CA1 Synapses: Correlation of Learning Performance with Pathway-Specific Plasticity"

_brainsci, 2024, doi:10.3390/brainsci14040382_

Round 1

Reviewer 1 Report

Comments and Suggestions for Authors

Authors examined developmental changes (betwen 16th and 23rd postnatal day) in training induced synaptic plasticity. It is continuation of their work on the developmental timing of synaptic plasticity. Contextual learning training induced a significant increase in the AMPA/NMDA ratio at CA3-CA1 synapses during postnatal days 16-23 and EC-CA1 synapses at postnatal days 16-21. They pinpoint postnatal day 23 as the mEPSCs and mIPSCs currents increase between P21 and P23. It is a very interesting contribution to the filed of developmental neurobiology.

The manuscript is clearly written and very well illustrated. The authors should introduce abbreviations in the abstract (PN, AMPA, NMDA), TTX should be expanded as it is used only once. The graphs on 3C and 3D lack the x-axis.

Author Response

We greatly appreciate the time and effort the reviewer has spent on our manuscript and the valuable comments provided. We are pleased to hear that our work is considered an interesting contribution to the field of developmental neurobiology. Following the reviewer's suggestions, we have made several revisions to our manuscript to address the concerns raised. Below, we respond to each point individually:

  1. Introduction of Abbreviations:As suggested, we have introduced and defined the abbreviations for postnatal day (PN), α-amino-3-hydroxy-5-methyl-4-isoxazolepropionic acid (AMPA), and N-methyl-D-aspartate (NMDA) in the abstract. We believe this will make our study more accessible to readers who are not familiar with these terms.
  2. Expansion of TTX: Tetrodotoxin (TTX) has now been expanded at its first use in the manuscript as recommended.
  3. Graphs on 3C and 3D: The x-axis adds untrained and trained.

Thank you again for considering our manuscript.

Reviewer 2 Report

Comments and Suggestions for Authors

This is an interesting/important/solid manuscript. In light of the recent attention to sexual dimorphism in neuroscience, the work would have been even stronger with female subjects. Perhaps the authors could comment on the usage of males in their rebuttal letter or in the method section (optional).

Minor: totally optional.

The responses of Schaffer and temporoammonic pathways appear strikingly similar. How does the AMPA/NMDA ratio of Schaffer vs. TA look like? Is it possible that the both fibers are stimulated? If possible, state the (typical) stimulus intensity.

In the abstract, the authors mention “temporal dynamics in (the) information entropy”. I did not understand what the authors meant by this phrase. Nor did I find an evaluation of PSC temporal dynamics. Perhaps the authors could clarify this in the discussion section. Could it be that the authors were referring to developmental dynamics?

Author Response

We are grateful for your constructive feedback on our manuscript and the opportunity to clarify and improve upon our work. Below, we address each comment in turn:

  1. Inclusion of Female Subjects: We recognize the importance of sexual dimorphism in neuroscience research and appreciate the reviewer's suggestion to include female subjects. The decision to use male subjects in the current study was based on our lab prior findings in males. However, we agree that including female subjects could provide valuable insights into the developmental timing of synaptic plasticity across sexes. Indeed, in our preliminary experiments, female rats showed shorter infantile amnesia than male rats, suggesting sex differences in the critical period for learning.We plan to explore potential sex differences in synaptic plasticity in our future studies.
  2. Similarity between Schaffer and Temporoammonic Pathways: The reviewer’s observation about the similarities between the responses of Schaffer collateral and temporoammonic (TA) pathways is insightful: there were no significant differences in the recorded waveforms in the previous report using 4-week-old animals(Paw-Min et al., Neuroscience 2000), in the current results using 2-3-week-old animals (Figure 4), and there was no difference in the absolute value of the AMPA/NMDA ratio. Since CA3-CA1 synapses are neuroanatomically much more abundant than ECIII-CA1 synapses, we agree that the lack of difference is more surprising. Unfortunately, we did not record differences in stimulus intensity, but there did not seem to be a significant difference.
  3. Clarification on Temporal Dynamics in Information Entropy: We apologize for any confusion caused by our use of the term “temporal dynamics in information entropy” in the abstract. This phrase was intended to describe the changes in synaptic efficacy and variability over developmental time. Upon review, we agree that this phrase may not have been clear to readers unfamiliar with our specific research context. We have edited this phrase.

    This revision should make the text more accessible and clarify our findings.We hope these responses and revisions address your concerns. We are thankful for the opportunity to improve our manuscript and believe these changes have enriched our study’s contributions to the field.